# L-SR1: A Second Order Optimization Method for Deep Learning

**Vivek Ramamurthy**
Sentient Technologies
1 California Street Suite 2300
San Francisco, CA 94111
`vivek.ramamurthy@sentient.ai`

**Nigel Duffy**
Sentient Technologies
1 California Street Suite 2300
San Francisco, CA 94111
`nigel.duffy@sentient.ai`

## Abstract

We describe L-SR1, a new second order method to train deep neural networks. Second order methods hold great promise for distributed training of deep networks. Unfortunately, they have not proven practical. Two significant barriers to their success are inappropriate handling of saddle points, and poor conditioning of the Hessian. L-SR1 is a practical second order method that addresses these concerns. We provide experimental results showing that L-SR1 performs at least as well as Nesterov's Accelerated Gradient Descent, on the MNIST and CIFAR10 datasets. For the CIFAR10 dataset, we see competitive performance on shallow networks like LeNet5, as well as on deeper networks like residual networks. Furthermore, we perform an experimental analysis of L-SR1 with respect to its hyperparameters to gain greater intuition. Finally, we outline the potential usefulness of L-SR1 in distributed training of deep neural networks.

## 1 Motivation

Second order methods hold great potential for distributing the training of deep neural networks. Due to their use of curvature information, they can often find good minima in far fewer steps than first order methods such as stochastic gradient descent (SGD). Moreover, stochastic second order methods can benefit from larger mini-batches (Le et al., 2011). This is because they estimate second derivatives via differences between estimated gradients. The gradient estimates need to have less variance, so that when we take their differences, the result has low variance. As a result they provide a different trade-off between number of steps and mini-batch size than do SGD-like methods. This trade-off is interesting, because while steps must be evaluated sequentially, a mini-batch may be evaluated in parallel. Thus, second order methods present an opportunity to extract more parallelism in neural network training. In particular, when mini-batches are sufficiently large, their evaluation may be distributed. Furthermore, there are relatively fewer hyperparameters to tune in second order methods, compared to variants of stochastic gradient descent.

L-BFGS (Nocedal, 1980; Liu & Nocedal, 1989) is perhaps the most commonly used second order method in machine learning. BFGS is a quasi-Newton method that maintains an approximation to the inverse Hessian of the function being optimized. L-BFGS is a limited memory version of BFGS that stores the most recent updates to the inverse Hessian approximation and can therefore be used practically for large scale problems. L-BFGS is typically combined with a line search technique to choose an appropriate step size at each iteration. L-BFGS has been used to good effect in convex optimization problems in machine learning, but has not found effective use in large scale non-convex problems such as deep learning.

Three critical weaknesses have been identified. First, we know that training deep neural networks involves minimizing non-convex error functions over continuous, high dimensional spaces. It has been argued that the proliferation of saddle points in these problems presents a deep and profound difficulty for quasi-Newton optimization methods (Dauphin et al., 2014). Furthermore, it has been argued that curvature matrices generated in second order methods are often ill-conditioned, and these need to be carefully repaired. A variety of approaches to this have been suggested, including the use of an empirical Fisher diagonal matrix (Martens, 2016). Finally, popular quasi-Newton

approaches, such as L-BFGS (in their default form), require line search to make parameter updates, which requires many more gradient and/or function evaluations.

We propose L-SR1, a second order method that addresses each of these concerns. SR1 (Symmetric Rank One) is a quasi-Newton method that uses a rank one update for updating the Hessian approximation of the function being optimized (Nocedal & Wright, 2006). Unlike BFGS, the SR1 update does not guarantee positive definiteness of the updated matrix. This was considered a major problem in the early days of nonlinear optimization when only line search iterations were used, and possibly led to the obscurity of SR1 outside the optimization community. However, with the development of trust-region methods, the SR1 updating formula is potentially very useful, and its ability to generate indefinite Hessian approximations can actually prove to be advantageous.

We believe that it is possible to overcome saddle points using rank-one update based second order methods. The more common rank-two methods, e.g. L-BFGS, maintain a positive definite approximation to the inverse of the Hessian, by design (Nocedal & Wright, 2006). At saddle-points, the true Hessian cannot be well approximated by a positive definite matrix, causing commonly used second order methods to go uphill (Dauphin et al., 2014). On the other hand, rank-one approaches such as SR1 don't maintain this invariant, so they can go downhill at saddle points. Numerical experiments (Conn et al., 1991) suggest that the approximate Hessian matrices generated by the SR1 method show faster progress towards the true Hessian than those generated by BFGS. This suggests that a limited memory SR1 method (L-SR1, if you like) could potentially outperform L-BFGS in the task of high dimensional optimization in neural network training. The building blocks needed to construct an L-SR1 method have been suggested in the literature (Byrd et al., 1994; Khalfan et al., 1993). To the best of our knowledge, however, there is no complete L-SR1 method previously described in the literature [1]. This prompted us to develop and test the approach, specifically in the large scale non-convex problems that arise in deep learning.

Two other insights make L-SR1 practical by removing the requirement for a line search and addressing the conditioning problem. First, we replace the line search using a trust region approach. While L-BFGS using line search is well studied, recently, an L-BFGS method that uses a trust-region framework has also been proposed (Burke et al., 2008). Second, we combine L-SR1 with batch normalization. Batch normalization is a technique of normalizing inputs to layers of a neural network, used to address a phenomenon known as *internal covariate shift* during training (Ioffe & Szegedy, 2015). Our hypothesis is that batch normalization may cause parameters of a neural network to be suitably scaled so that the Hessian becomes better conditioned. We tested this hypothesis empirically and outline the results below.

## 2 RELATED WORK

We now briefly summarize some other second order approaches that have been suggested in the literature, in order to place our approach in context. Pearlmutter (1994) derived a technique that directly calculated the product of the Hessian with an arbitrary vector, and applied this technique to a few variants of backpropagation, thereby showing a way to use the full Hessian without needing to compute and store it. Martens (2010) used a generalization of this technique, introduced by Schraudolph (2002), to develop a second order optimization method based on the "Hessian-free" approach, using it to train deep auto-encoders (Martens, 2010), as well as recurrent neural networks (Martens & Sutskever, 2011). The "Hessian-free" approach is essentially a line search Newton-CG (Conjugate Gradient) method, also known as the truncated Newton method (Nocedal & Wright, 2006), in which the search direction is computed by applying CG to the Newton method, and terminating it once it has made sufficient progress. This approach differs from ours in its use of line search instead of a trust region method. Moreover, it computes Hessian-vector products using finite differencing, as opposed to the limited-memory symmetric rank one update with trust region method, used in our approach. The cost of skipping the Hessian calculation in a truncated Newton method is one additional gradient evaluation per CG iteration (Nocedal & Wright, 2006). As mentioned previously, Dauphin et al. (2014) argue, that in high dimensional problems of practical interest, the proliferation of saddle points poses greater difficulty than local minima. In a bid to escape these saddle points, they propose second order optimization method called the saddle-free Newton method. Key to this

---

[1] The reference Brust et al. (2016) describes an approach to solve the trust region sub-problem encountered in an L-SR1 method, but does not describe the L-SR1 method itself.

approach is the definition of a class of generalized trust region methods. This class extends classical trust region methods in a couple of ways. A first order Taylor expansion of the function is minimized, instead of the second order Taylor expansion. Moreover, the constraint on the step norm is replaced by generalized constraint on the distance between consecutive iterates. Our approach, by contrast, uses a a classical trust-region method. Rather than compute the Hessian exactly, Dauphin et al. (2014) use an approach similar Krylov subspace descent (Vinyals & Povey, 2012). The function is optimized in a lower-dimensional Krylov subspace, which is determined through Lanczos iteration of the Hessian (Vinyals & Povey, 2012). The Lanczos method may be considered a generalization of the CG method that can be applied to indefinite systems, and may be used to aid the CG method by gathering negative curvature information (Nocedal & Wright, 2006). The Lanczos method also involves finding an approximate solution to a trust-region subproblem in the range of a Krylov basis that it generates. This trust region problem differs from the one we solve, in that the Krylov basis generated has a special structure due to its mapping to a tridiagonal matrix (Nocedal & Wright, 2006).

It is worth noting that several approaches have been proposed to overcome the weaknesses of L-BFGS. First, it has been proposed to initialize L-BFGS with a number of SGD steps. However, this diminishes the potential for parallelism (Dean et al., 2012; Le et al., 2011). Second, it has been proposed to use "forgetting", where every few (say, for example, 5) steps, the history for L-BFGS is discarded. However, this greatly reduces the ability to use second order curvature information. There has also been a recent spurt of work on stochastic quasi-Newton methods for optimization. Byrd et al. (2016) propose a stochastic quasi-Newton method which uses the classical L-BFGS formula, but collects curvature information pointwise, at regular intervals, through sub-sampled Hessian vector products, rather than at every iteration. Mokhtari & Ribeiro (2014) propose RES, a regularized stochastic version of BFGS to solve convex optimization problems with stochastic objectives, and prove its convergence for bounded Hessian eigenvalues. Mokhtari & Ribeiro (2015) propose an online L-BFGS method for solving optimization problems with strongly convex stochastic objectives, and establish global almost sure convergence of their approach for bounded Hessian eigenvalues of sample functions. In the case of nonconvex stochastic optimization, Wang et al. (2014) propose, based on a general framework, two concrete stochastic quasi-Newton update strategies, namely stochastic damped-BFGS update and stochastic cyclic Barzilai-Borwein-like update, to adaptively generate positive definite Hessian approximations. They also analyze the almost sure convergence of these updates to stationary points. Keskar & Berahas (2015) propose ADAQN, a stochastic quasi-Newton algorithm for training RNNs. This approach retains a low per-iteration cost while allowing for non-diagonal scaling through a stochastic L-BFGS updating scheme. The method also uses a novel L-BFGS scaling initialization scheme and is judicious in storing and retaining L-BFGS curvature pairs. Finally, Curtis (2016) proposes a variable-metric algorithm for stochastic nonconvex optimization which exploits fundamental self-correcting properties of BFGS-type updating, and uses it to solve a few machine learning problems. As one may notice, all of these approaches adapt the BFGS-style rank two updates in different ways to solve convex and non-convex problems. In contrast, our approach uses SR1-type updates, which we think can help better navigate the pathological saddle points present in the non-convex loss functions found in deep learning, by not constraining the Hessian approximation to be positive definite, as in the case of BFGS-style updates. Comparison of our approach with one of these recent stochastic second order methods is an interesting next step. In the Appendix, we provide a brief primer on line search and trust region methods, as well as on quasi-Newton methods and their limited memory variants.

## 3   Algorithm

Our algorithm is synthesized as follows. We take the basic SR1 algorithm described in Nocedal & Wright (2006) (Algorithm 6.2), and represent the relevant input matrices using the limited-memory representations described in Byrd et al. (1994). The particular limited-memory representations used in the algorithm vary, depending on whether we use trust region or line search methods as subroutines to make parameter updates, as does some of the internal logic. For instance, if $k$ updates are made to the symmetric matrix $B_0$ using the vector pairs $\{s_i, y_i\}_{i=0}^{k-1}$ and the SR1 formula, the resulting matrix $B_k$ can be expressed as (Nocedal & Wright, 2006)

$$B_k = B_0 + (Y_k - B_0 S_k)(D_k + L_k + L_k^T - S_k^T B_0 S_k)^{-1}(Y_k - B_0 S_k)^T$$

where $S_k$, $Y_k$, $D_k$, and $L_k$ are defined as follows:

$$S_k = [s_o, \cdots, s_{k-1}], and Y_k = [y_0, \cdots, y_{k-1}]$$

,

$$(L_k)_{i,j} = \begin{cases} s_{i-1}^T y_{j-1} & \text{if } i > j \\ 0 & \text{otherwise} \end{cases}$$

$$D_k = \text{diag}[s_0^T y_0, \cdots, s_{k-1}^T y_{k-1}]$$

The self-duality of the SR1 method (Nocedal & Wright, 2006) allows the inverse formula $H_k$ to be obtained simply by replacing $B$, $s$, and $y$ by $H$, $y$, and $s$, respectively, using standard matrix identities. Limited-memory SR1 methods can be derived exactly like in the case of the BFGS method. Additional details are present in the pseudocode provided in the Appendix. The algorithm we develop is general enough to work with any line search or trust region method. While we tested the algorithm with line search approaches described in Dennis Jr. & Schnabel (1983), and with the trust region approach described in Brust et al. (2016), in this paper, we focus our experimental investigations on using the trust region approach, and the advantage that provides over using other first and second order optimization methods.

We also make a note here about the space and time complexity of our algorithm. We respectively denote by $m$ and $n$, the memory size, and parameter dimensions. We assume $m << n$. As discussed in Section 7.2 of Nocedal & Wright (2006), the limited-memory updating procedure of $B_k$ requires approximately $2mn + O(m^3)$ operations, and matrix vector products of the form $B_k v$ can be performed at a cost of $(4m + 1)n + O(m^2)$ multiplications. Moreover, the Cholesky and eigenvalue decompositions we perform within our trust-region method for $m \times m$ matrices require $O(m^3)$ operations. It follows quite easily[2] from this that the space complexity of our algorithm is $O(mn)$, and the per iteration time complexity of our algorithm is $O(mn)$.

## 4 EXPERIMENTS

In the following, we summarize the results of training standard neural networks on the MNIST and CIFAR10 datasets using our approach, and benchmarking the performance with respect to other first and second order methods. First, we compared our L-SR1 (with trust region) approach, with Nesterov's Accelerated Gradient Descent (NAG), L-BFGS with forgetting every 5 steps, default SGD, AdaDelta, and SGD with momentum, by training small standard networks on the MNIST and CIFAR10 datasets. On these problems, we also studied the effect of varying the minibatch size, for L-SR1, Adam (Kingma & Ba, 2014), and NAG. Next, we compared our L-SR1 with trust region approach with default hyperparameters, with a benchmark SGD with momentum, and Adam, by training a 20-layer deep residual network on the CIFAR10 dataset. Following that, we varied each hyperparameter of the L-SR1 with trust region approach to observe its effect on training the residual network on CIFAR10.

### 4.1 LENET-LIKE NETWORKS

For each approach, and for each dataset, we considered the case where our networks had batch normalization layers within them, and the case where they did not. The parameters of the networks were randomly initialized. All experiments were repeated 10 times to generate error bars.

### 4.1.1 MNIST

We considered the LeNet5 architecture in this case, which comprised 2 convolutional layers, followed by a fully connected layer and an outer output layer. Each convolutional layer was followed by a max-pooling layer. In the case where we used batch-normalization, each convolutional and fully connected layer was followed by a spatial batch normalization layer. We used a mini-batch size of 20 for the first order methods like NAG, SGD, AdaDelta and SGD with momentum, and a mini-batch size of 400 for the second order methods like L-SR1 and L-BFGS. The memory size was set to 5 for both L-SR1 and L-BFGS. The networks were trained for 20 epochs. Further details on the network architecture and other parameter settings are provided in the Appendix.

---

[2]Deep neural networks typically have paramater dimensions in the tens of millions, while the memory size typically does not exceed 10. So $n$ is indeed several orders of magnitude larger than $m$.

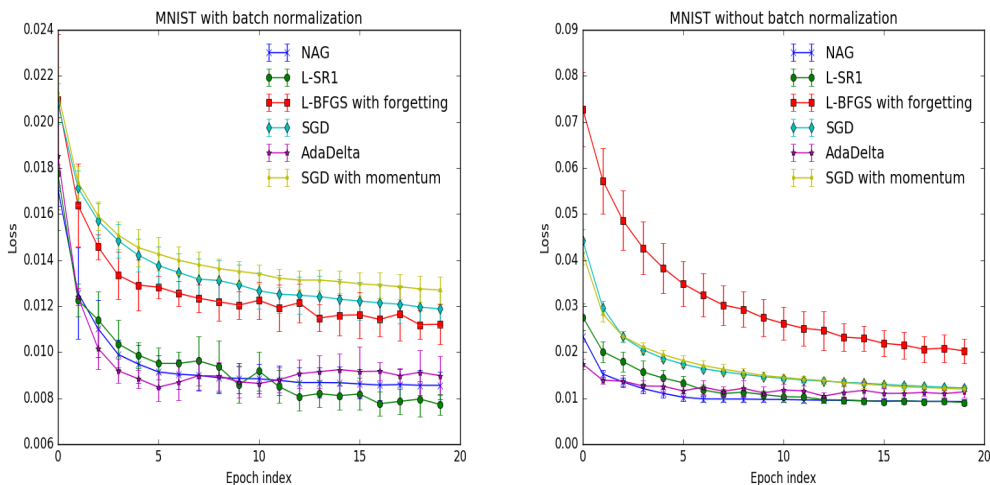

Figure 1: Variation of test loss with number of epochs, on the MNIST dataset, with and without batch normalization. Note that the scales on the y-axes are different.

### 4.1.2 CIFAR10

We considered a slight modification to the 'LeNet5' architecture described above. We used a mini-batch size of 96 for NAG, SGD, AdaDelta and SGD with momentum. The other mini-batch sizes and memory sizes for L-SR1 and L-BFGS were as above. As above, the networks were trained for 20 epochs. Further details on the network architecture and other parameter settings are provided in the Appendix.

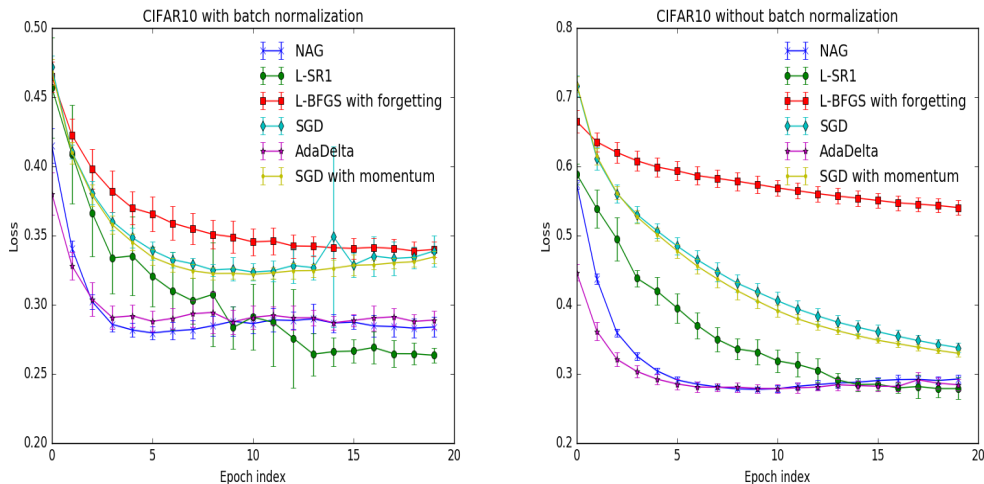

Figure 2: Variation of test loss with number of epochs, on the CIFAR10 dataset, with and without batch normalization. Note that the scales on the y-axes are different.

### 4.1.3 VARIATION OF MINIBATCH SIZE

We also compared the variation of test loss between L-SR1, Adam and NAG, as we varied the mini-batch size from 500 to 1000 to 10000, in the presence of batch normalization. The network architectures were as above. For minibatch sizes 500 and 1000, we trained the networks for 50 epochs, while for the minibatch size of 10000, the networks were trained for 200 epochs.

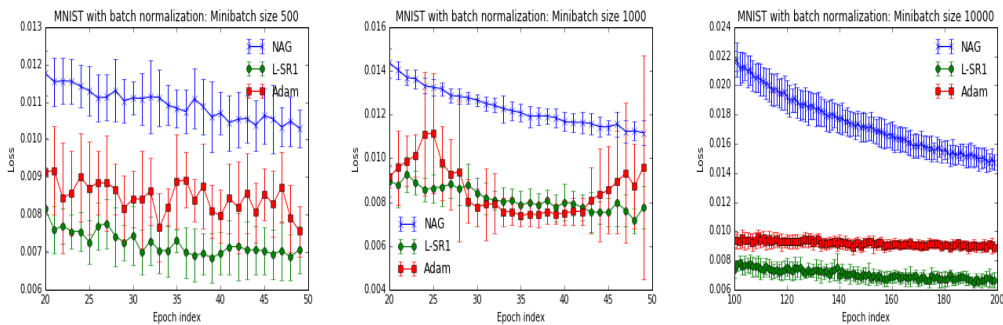

Figure 3: Variation of test loss with number of epochs, on the MNIST dataset, with batch normalization, for varying minibatch sizes. Note that the scales on the x and y-axes across figures are different.

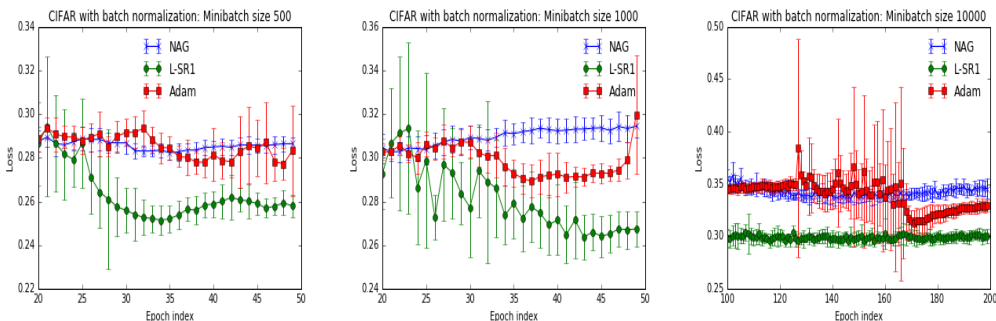

Figure 4: Variation of test loss with number of epochs, on the CIFAR10 dataset, with batch normalization, for varying minibatch sizes. Note that the scales on the x and y-axes across figures are different.

### 4.1.4 DISCUSSION

Our first set of experiments (Figures 1, 2) suggest that L-SR1 performs as well as, or slightly better than all the first order methods on both the MNIST and CIFAR10 datasets, with or without batch normalization. L-SR1 is substantially better than L-BFGS in all settings, with or without forgetting. Forgetting appears to be necessary in order to get L-BFGS to work. Without forgetting, the approach appears to be stuck where it is initialized. For this reason, the plots for L-BFGS without forgetting have not been included. Batch normalization appears to improve the performance of all approaches, particularly the early performance of second order approaches like L-SR1 and L-BFGS.

The experiments with variation of minibatch sizes (Figures 3, 4), seem to provide compelling evidence of the potential for distributed training of deep networks, as may be seen from Table 1. First, we note that first order methods like NAG are not as sensitive to size of the minibatch, as commonly understood. For example, a 20 fold increase in minibatch size did not decrease the speed of convergence by the same or higher order of magnitude. Furthermore, approaches like L-SR1 and Adam appear to be much less sensitive to increasing minibatch size than NAG. This strengthens the case for their application to distributed training of deep neural networks. Finally, while Adam makes much faster initial progress than the other approaches, its final test loss by the end of training is worse than in the case of L-SR1.

One of the limitations of SR1 updating is that the denominator in the update can vanish. The literature however suggests that this happens rarely enough that the updates can be skipped when this phenomenon occurs, without affecting performance. In this regard, we had some interesting observations from our experiments. While in most cases, updates were either never skipped, or skipped less than 2.5% of the time, the cases of MNIST training with batch normalization, yielded abnor-

| | Number of epochs needed to surpass target test loss | | | | | |
|---|---|---|---|---|---|---|
| | MNIST with batch normalization | | | CIFAR10 with batch normalization | | |
| Target test loss (%) | 1.1 | 1.2 | 1.5 | 30 | 31 | 34 |
| Minibatch size | 500 | 1000 | 10000 | 500 | 1000 | 10000 |
| NAG | 35 | 37 | 194 | 9 | 17 | 125 |
| L-SR1 | 5 | 6 | 8 | 16 | 14 | 41 |
| Adam | 4 | 3 | 10 | 6 | 6 | 15 |

Table 1: Speed of convergence of NAG, L-SR1, and Adam, with varying minibatch sizes.

mally high levels of skipped updates, ranging all the way from 7% to higher than 60% (for minibatch size 10000). While this did not seem to affect performance adversely, it certainly warrants future investigation. Moreover, a better understanding of the interplay between batch normalization and optimization could help inform potential improvements in optimization approaches.

## 4.2 RESIDUAL NETWORKS

We next considered a deeper residual network architecture described in section 4.2 of He et al. (2015b), with $n = 3$. This led to a 20-layer residual network including 9 shortcut connections. As in He et al. (2015b), we used batch normalization (Ioffe & Szegedy, 2015) and the same initialization method (He et al., 2015a).

### 4.2.1 COMPARISON WITH SGD WITH MOMENTUM, AND ADAM

We trained the residual network using the benchmark SGD with momentum, and other parameter settings as described in He et al. (2015b). We also trained the network using L-SR1 with default settings. These included, a memory size of 5, a trust-region radius decrease factor of 0.5, and a trust-region radius increase factor of 2.0. Finally, we also compared with Adam, with default settings (Kingma & Ba, 2014). We used the same mini-batch size of 128 for all algorithms. Based on the learning rate schedule used, the learning rate was equal to 0.1 through the first 80 epochs, 0.01 up to 120 epochs, and 0.001 thereafter, for SGD with momentum. Figure 5 shows variation of test loss, over epochs, and by time. It needs to be noted that default L-SR1, with no parameter tuning at all, has a superior final test loss to Adam, and is competitive with SGD with momentum, which used custom parameters that were tuned carefully. L-SR1 does make slower progress over time, which can be further optimized. Finally, we note that the test loss for L-SR1 bounces around a lot more than the test loss for the other algorithms. This bears further exploration.

### 4.2.2 VARIATION OF L-SR1 HYPERPARAMETERS

We varied the hyperparameters of L-SR1 in turn, keeping the remaining fixed. In each case, we trained the network for 200 epochs. We first considered varying the increase and decrease factors together. We considered a trust-region radius decrease factor of 0.2, 0.5 and 0.8, and a trust-region radius increase factor 1.2 and 2.0. The respective default values of these factors are 0.5 and 2.0 respectively. This led to six different combinations of decrease and increase factors. We kept the memory size and mini-batch size fixed at 5 and 128 respectively. Next, we considered memory sizes of 2 and 10 (in addition to 5, which we tried earlier), keeping the mini-batch size, decrease factor, and increase factor fixed at 128, 0.5, and 2.0 respectively. Finally, we considered mini-batch sizes of 512, 2048 and 8192 (in addition to 128, which we tried earlier), keeping the memory size, decrease factor, and increase factor fixed at 5, 0.5, and 2.0 respectively. Figure 6 shows the results.

The following may be noted, based on the experiments with L-SR1 for training a residual network on CIFAR10. While there is potential value in increasing and decreasing the trust region radius at different rates, our experiments suggest that it may not be necessary to tune these hyperparameters. There is no noticeable performance gain from using a higher memory size in L-SR1. Furthermore, using a smaller memory size performs at least as well as in the default case. This is good news, due to the consequent savings in storage and computational resources. L-SR1 is relatively insensitive to a 4-fold increase in mini-batch size from 128 to 512, and a further 4-fold increase to 2048. The minibatch sensitivity of L-SR1 seems to be higher in the case of the residual network, compared

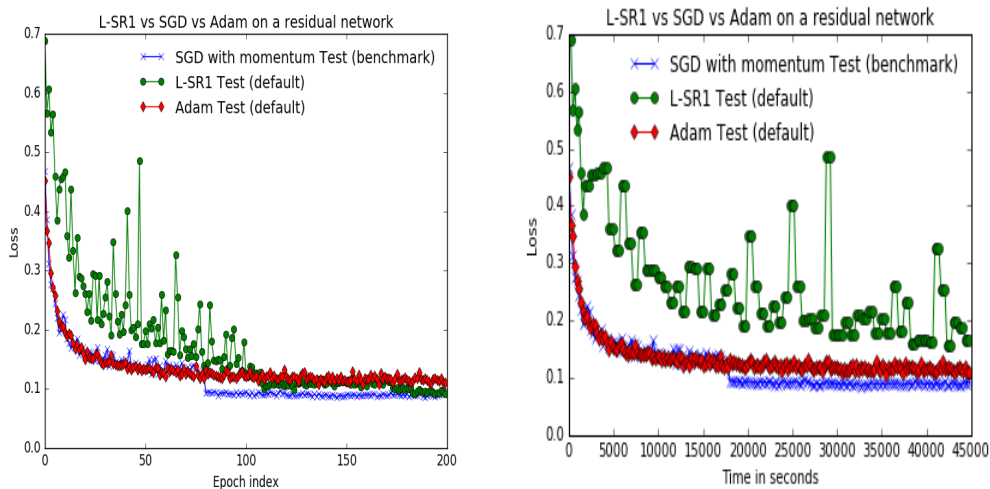

Figure 5: LSR1 vs SGD vs Adam, on the CIFAR10 dataset, using a residual network. The x-axis on the left shows number of epochs, while the x-axis on the right shows time in seconds.

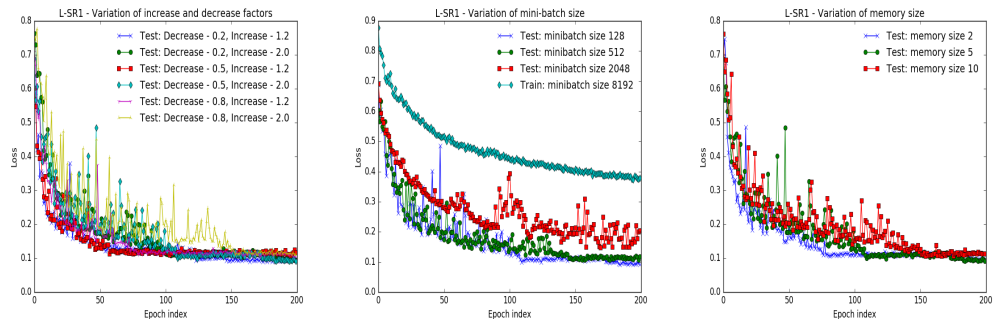

Figure 6: Variation of trust region radius increase and decrease factors, mini-batch size and memory size with number of epochs, on the CIFAR10 dataset, using a residual network. Note that the scales on the y-axes are different.

with the Le-Net like networks seen earlier. Finally, we found the proportion of skipped updates in the case of residual networks to be less than $0.5\%$ in all cases.

## 5 CONCLUSIONS

In this paper, we have described L-SR1, a new second order method to train deep neural networks. Our experiments suggest that this approach is at the very least, competitive, with other first order methods, and substantially better than L-BFGS, a well-known second order method. Our experiments also appear to validate our intuition about the ability of L-SR1 to overcome key challenges associated with second order methods, such as inappropriate handling of saddle points, and poor conditioning of the Hessian. Our experimentation with the hyperparameters of L-SR1 suggested that it is relatively robust with respect to them, and requires minimal tuning. Furthermore, we have evidence to suggest that L-SR1 is much more insensitive to larger minibatch sizes than a first order method like NAG. This suggests that L-SR1 holds promise for distributed training of deep networks, and we see our work as an important step toward that goal.

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

## APPENDIX

### BACKGROUND

In the following, we provide a brief primer on line search and trust region methods, as well as on quasi-Newton methods and their limited memory variants. Further details may be found in Nocedal & Wright (2006).

### LINE SEARCH AND TRUST REGION METHODS

In any optimization algorithm, there are two main ways of moving from the current point $x_k$ to a new iterate $x_{k+1}$. One of them is line search. In it, the algorithm picks a descent direction $p_k$ and searches along this direction from the current iterate $x_k$ for a new iterate with a lower function value. The distance $\alpha$ to move along $p_k$ can be found by solving the following one-dimensional minimization problem:

$$\min_{\alpha>0} f(x_k + \alpha p_k)$$

Instead of an exact minimization which may be expensive, the line search algorithm generates a limited number of trial step lengths until it finds one that generates a sufficient decrease in function

value. At the new point, the process of computing the descent direction and step length is repeated. The other way is to use a trust region method. In a trust region method, the information about $f$ is used to construct a model function $m_k$, which is supposed to approximate $f$ near the current point $x_k$. Since the model $m_k$ may not approximate $f$ well when $x$ is far from $x_k$, the search for a minimizer of $m_k$ is restricted to some trust region within a radius $\Delta_k$ around $x_k$. To wit, the candidate step $p$ approximately solves the following sub-problem:

$$\min_{p:\|p\|\leq\Delta_k} m_k(x_k + p),$$

If the candidate solution does not produce a sufficient decrease in $f$, the trust region is considered too large for the model function to approximate $f$ well. So we shrink the trust region and re-solve. Essentially, the line search and trust region approaches differ in the order in which they choose the direction and magnitude of the move to the next iterate. In line search, the descent direction $p_k$ is fixed first, and then the step length $\alpha_k$ to be taken along that direction is computed. In trust region, a maximum distance equal to the trust-region radius $\Delta_k$ is first set, and then a direction is determined within this radius, that achieves the best improvement in the objective value. If such a direction does not yield sufficient improvement, the model function is determined to be a poor approximation to the function, and the trust-region radius $\Delta_k$ is reduced until the approximation is deemed good enough. Conversely, as long as the model function appears to approximate the objective function well, the trust region radius is increased until the approximation is not good enough.

LIMITED MEMORY QUASI-NEWTON METHODS

Quasi-Newton methods are a useful alternative to Newton's method in that they do not require computation of the exact Hessian, and yet still attain good convergence. In place of the true Hessian $\nabla^2 f_k$, they use an approximation $B_k$, which is updated after each step based on information gained during the step. At each step, the new Hessian approximation $B_{k+1}$ is required to satisfy the following condition, known as the *secant equation*:

$$B_{k+1}s_k = y_k$$

where

$$s_k = x_{k+1} - x_k, y_k = \nabla f_{k+1} - \nabla f_k$$

Typically, $B_{k+1}$, is also required to be symmetric (like the exact Hessian), and the difference between successive approximations $B_k$ and $B_{k+1}$ is constrained to have low rank. One of the most popular formulae for updating the Hessian approximation $B_k$ is the *BFGS formula*, named after its inventors, Broyden, Fletcher, Goldfarb, and Shanno, which is defined by

$$B_{k+1} = B_k - \frac{B_k s_k s_k^T B_k}{s_k^T B_k s_k} + \frac{y_k y_k^T}{y_k^T s_k}$$

A less well known formula, particularly in the machine learning community, is the *symmetric-rank-one* (SR1) formula, defined by

$$B_{k+1} = B_k + \frac{(y_k - B_k s_k)(y_k - B_k s_k)^T}{(y_k - B_k s_k)^T s_k}$$

The former update is a rank-two update, while the latter is a rank-one update. Both updates satisfy the secant equation and maintain symmetry. The BFGS update always generates positive definite approximations whenever the initial approximation $B_0$ is positive definite and $s_k^T y_k > 0$. Often, in practical implementations of quasi-Newton methods, the inverse Hessian approximation $H_k$ is used instead of the $B_k$, and the corresponding update formulae can be generated using the Sherman-Morrison-Woodbury matrix identity (Hager, 1989).

Limited-memory quasi-Newton methods are useful for solving large problems where computation of Hessian matrices is costly or when these matrices are dense. Instead of storing fully dense $n \times n$ approximations, these methods save only a few vectors of length $n$ that capture the approximations. Despite these modest storage requirements, they often converge well. The most popular limited memory quasi-Newton method is L-BFGS, which uses curvature information from only the most recent iterations to construct the inverse Hessian approximation. Curvature information from earlier

iterations, which is less likely to be useful to modeling the actual behavior of the Hessian at the current iteration, is discarded in order to save memory.

Limited-memory quasi-Newton approximations can be used with line search or trust region methods. As described in Byrd et al. (1994), we can derive efficient limited memory implementations of several quasi-Newton update formulae, and their inverses.

NETWORK ARCHITECTURES AND HYPERPARAMETER SETTINGS

MNIST

The layers of the LeNet5 architecture used, are described below. All the batch normalization layers were removed, in the 'without batch normalization' case.

- Convolutional Layer - filter size $5 \times 5$, 20 feature maps, stride 1, padding 0, and a ReLU activation function with bias 0 and Gaussian noise with mean 0 and standard deviation 0.1
- Spatial Batch Normalization Layer
- Max Pooling Layer - filter size 2
- Convolutional Layer - filter size $5 \times 5$, 50 feature maps, stride 1, padding 0, and a ReLU activation function with bias 0 and Gaussian noise with mean 0 and standard deviation 0.1
- Spatial Batch Normalization Layer
- Max Pooling Layer - filter size 2
- Fully Connected Layer - 500 hidden units, and a tangent hyperbolic activation function
- Spatial Batch Normalization Layer
- Outer Output Layer - 10 outputs and output standard deviation of 0.1

Additionally, the network was trained with $L2$ regularization with parameter 0.0001. Training loss was measured as softmax cross entropy, while test loss was measured as multi-class error count. In the case of the first order methods, the learning rate was set to 0.003 where needed, and the momentum was set to 0.9, where needed. AdaDelta did not take any parameters.

CIFAR10

The layers of the architecture used, are described below. All the batch normalization layers were removed, in the 'without batch normalization' case.

- Convolutional Layer - filter size $5 \times 5$, 32 feature maps, stride 1, padding 2, and a ReLU activation function with bias 0 and Gaussian noise with mean 0 and standard deviation 0.01
- Spatial Batch Normalization Layer
- Max Pooling Layer - filter size 2
- Activation Layer - ReLU activation function with bias 0 and Gaussian noise with mean 0 and standard deviation 0.1
- Convolutional Layer - filter size $5 \times 5$, 32 feature maps, stride 1, padding 2, and a ReLU activation function with bias 0 and Gaussian noise with mean 0 and standard deviation 0.01
- Spatial Batch Normalization Layer
- Max Pooling Layer - filter size 2
- Convolutional Layer - filter size $5 \times 5$, 64 feature maps, stride 1, padding 2, and a ReLU activation function with bias 0 and Gaussian noise with mean 0 and standard deviation 0.01
- Spatial Batch Normalization Layer
- Max Pooling Layer - filter size 2
- Fully Connected Layer - 64 hidden units, and a ReLU activation function with bias 0 and Gaussian noise with mean 0 and standard deviation 0.1
- Spatial Batch Normalization Layer

- Outer Output Layer - 10 outputs and output standard deviation of $0.1$

Additionally, the network was trained with $L2$ regularization with parameter $0.001$. Training loss was measured as softmax cross entropy, while test loss was measured as multi-class error count. In the case of the first order methods, the learning rate was set to $0.01$ where needed, and the momentum was set to $0.9$, where needed. AdaDelta did not take any parameters.

PSEUDOCODE

Algorithm 1 provides the pseudocode for L-SR1 with trust region method, while Algorithm 2 provides the pseudocode for L-SR1 with line search.

---

**Algorithm 1** L-SR1 with Trust Region Method

---

**Require:** $S_k = [s_0, \cdots, s_{k-1}]$, $Y_k = [y_0, \cdots, y_{k-1}]$, starting point $x_0 \in \mathbb{R}^n$, limited memory size $m << n$, initial Hessian approximation $B_0$ (a diagonal matrix, typically $\gamma I_n$, $\gamma \neq 0$), initial trust-region radius $\Delta_0 = \|\nabla f(x_0)\|_2$, convergence tolerance $t > 0$, maximum iterations $K$, parameters $\eta \in (0, 10^{-3})$, $r \in (0, 1)$, and column dimension $colDim$;

1: $k \leftarrow 0$
2: **while** $k < K$ and $\|\nabla f(x_k)\|_2 > t$ and $\|s_k\|_2 > t$ **do**
3: **if** $k = 0$ or $S_k.colDim = 0$ **then**
4: $s_k \leftarrow -\nabla f(x_k)$
5: $B_k \leftarrow B_0$
6: **else**
7: $s_k \leftarrow$ TrustRegionMethod$(\Psi_k, M_k^{-1}, \nabla f(x_k), \Delta_k, \gamma, B_0)$ (Solve the trust-region sub-problem)
8: $B_k s_k \leftarrow B_0 s_k + \Psi_k M_k(\Psi_k^T s_k)$
9: **end if**
10: $pred \leftarrow -\left(\nabla f(x_k)^T s_k + \frac{1}{2} s_k^T B_k s_k\right)$ (predicted reduction)
11: $ared \leftarrow f(x_k) - f(x_k + s_k)$ (actual reduction)
12: $y_k \leftarrow \nabla f(x_k + s_k) - \nabla f(x_k)$
13: **if** $ared/pred > \eta$ **then**
14: $x_{k+1} \leftarrow x_k + s_k$
15: **else**
16: $x_{k+1} \leftarrow x_k$
17: **end if**
18: $\Delta_{k+1} \leftarrow \Delta_k$
19: **if** $ared/pred > u$ **then** ($u = 0.75$ by default)
20: **if** $\|s_k\| > \rho\Delta_k$ **then** ($\rho = 0.8$ by default)
21: $\Delta_{k+1} \leftarrow 2\Delta_k$
22: **end if**
23: **else if** $ared/pred < l$ **then** ($l = 0.1$ by default)
24: $\Delta_{k+1} \leftarrow 0.5\Delta_k$
25: **end if**
26: **if** $|s_k^T(y_k - B_k s_k)| \geq r\|s_k\|\|y_k - B_k s_k\|$ **then**
27: $S_{k+1} \leftarrow [S_k, s_k]$, $Y_{k+1} \leftarrow [Y_k, y_k]$
28: **if** $S_{k+1}.colDim > m$ **then**
29: $S_{k+1} \leftarrow S_{k+1}[, 2 : m + 1]$, $Y_{k+1} \leftarrow Y_{k+1}[, 2 : m + 1]$
30: **end if**
31: **while** $S_{k+1}.colDim > 0$ **do**
32: $\Psi_{k+1} \leftarrow Y_{k+1} - B_0 S_{k+1}$
33: $M_{k+1}^{-1} \leftarrow S_{k+1}^T Y_{k+1} - S_{k+1}^T B_0 S_{k+1}$
34: **if** $(\Psi_{k+1}^T \Psi_{k+1} \succ 0$ and $|M_{k+1}^{-1}| \neq 0)$ **then**
35: **break**
36: **else**
37: Remove the first columns of $S_{k+1}$ and $Y_{k+1}$
38: **end if**
39: **end while**
40: **end if**
41: $k \leftarrow k + 1$
42: **end while**

---

---

**Algorithm 2** L-SR1 with Line Search

---

**Require:** $S_k = [s_0, \cdots, s_{k-1}]$, $Y_k = [y_0, \cdots, y_{k-1}]$, starting point $x_0 \in \mathbb{R}^n$, limited memory size $m << n$, initial inverse Hessian approximation $H_0$ (a diagonal matrix, typically $I_n$), initial step length $\lambda_0 = 1$, convergence tolerance $t > 0$, maximum iterations $K$, $r \in (0, 1)$, and column dimension $colDim$;

1: $k \leftarrow 0$
2: **while** $k < K$ and $(\|\nabla f(x_k)\|_2 > t$ or $\|s_k\|_2 > t)$ **do**
3:       **if** $k = 0$ or $S_k.colDim = 0$ **then**
4:             $d_k \leftarrow -\nabla f(x_k)$
5:             $H_k \leftarrow H_0$
6:       **else**
7:             $d_k \leftarrow -H_0 \nabla f(x_k) - \Psi_k M_k (\Psi_k^T \nabla f(x_k))$
8:       **end if**
9:       **if** $k > 0$ **then**
10:             $\lambda_0 = \min\{1, 2\lambda_{k-1}\}$
11:       **end if**
12:       $\lambda_k \leftarrow$ computeStepLength$(f, x_k, d_k, \lambda_0)$ (perform line search)
13:       $s_k \leftarrow \lambda_k d_k$
14:       $x_{k+1} \leftarrow x_k + s_k$
15:       $y_k \leftarrow \nabla f(x_{k+1}) - \nabla f(x_k)$
16:       **if** $k > 0$ and $S_k.colDim > 0$ **then**
17:             $H_k y_k \leftarrow H_0 y_k + \Psi_k M_k (\Psi_k^T y_k)$
18:       **end if**
19:       **if** $|y_k^T (s_k - H_k y_k)| \geq r \|y_k\| \|s_k - H_k y_k\|$ **then**
20:             $S_{k+1} \leftarrow [S_k, s_k]$, $Y_{k+1} \leftarrow [Y_k, y_k]$
21:             **if** $S_{k+1}.colDim > m$ **then**
22:                   $S_{k+1} \leftarrow S_{k+1}[, 2 : m + 1]$, $Y_{k+1} \leftarrow Y_{k+1}[, 2 : m + 1]$
23:             **end if**
24:             **while** $S_{k+1}.colDim > 0$ **do**
25:                   $\Psi_{k+1} \leftarrow S_{k+1} - H_0 Y_{k+1}$
26:                   $M_{k+1}^{-1} \leftarrow Y_{k+1}^T S_{k+1} - Y_{k+1}^T H_0 Y_{k+1}$
27:                   **if** $(|M_{k+1}^{-1}| \neq 0)$ **then**
28:                         **break**
29:                   **else**
30:                         Remove the first columns of $S_{k+1}$ and $Y_{k+1}$
31:                   **end if**
32:             **end while**
33:       **end if**
34:       $k \leftarrow k + 1$
35: **end while**

---

