# Peer review of "L-SR1: A Second Order Optimization Method for Deep Learning"

_ICLR 2017 — rejected_

[Public Comment · (anonymous) · 19 Nov 2016]
**A few comments**

This is a very interesting paper and the results look interesting. The paper is missing references for a large body of stochastic quasi-Newton methods and at times, is a bit misleading with its claims. 

References:
A Self-Correcting Variable-Metric Algorithm for Stochastic Optimization (ICML, 2016):

[Official Review · AnonReviewer2 · rating 5 · confidence 3 · 14 Dec 2016]
**Address better optimization at saddle points with symmetric rank-one method which does not guarantee pos. def. update matrix, vs. BFGS approach. Investigating this optimization with limited memory version or SR1**

It is an interesting idea to go after saddle points in the optimization with an SR1 update and a good start in experiments, but missing important comparisons to recent 2nd order optimizations such as Adam, other Hessian free methods (Martens 2012), Pearlmutter fast exact multiplication by the Hessian. From the mnist/cifar curves it is not really showing an advantage to AdaDelta/Nag (although this is stated), and much more experimentation is needed to make a claim about mini-batch insensitivity to performance, can you show error rates on a larger scale task?

[Official Review · AnonReviewer1 · rating 4 · confidence 4 · 16 Dec 2016]
**Interesting work, but not ready to be published**

The paper proposes a new second-order method L-SR1 to train deep neural networks. It is claimed that the method addresses two important optimization problems in this setting: poor conditioning of the Hessian and proliferation of saddle points. The method can be viewed as a concatenation of SR1 algorithm of Nocedal & Wright (2006) and limited-memory representations Byrd et al. (1994). First of all, I am missing a more formal, theoretical argument in this work (in general providing more intuition would be helpful too), which instead is provided in the works of Dauphin (2014) or Martens. The experimental section in not very convincing considering that the performance in terms of the wall-clock time is not reported and the advantage over some competitor methods is not very strong even in terms of epochs. I understand that the authors are optimizing their implementation still, but the question is: considering the experiments are not convincing, why would anybody bother to implement L-SR1 to train their deep models? The work is not ready to be published.

[Official Review · AnonReviewer3 · rating 4 · confidence 3 · 16 Dec 2016]
**O(mn)?**

L-SR1 seems to have O(mn) time complexity. I miss this information in your paper. 
Your experimental results suggest that L-SR1 does not outperform Adadelta (I suppose the same for Adam). 
Given the time complexity of L-SR1, the x-axis showing time would suggest that L-SR1 is much (say, m times) slower. 
"The memory size of 2 had the lowest minimum test loss over 90" suggests that the main driven force of L-SR1 
was its momentum, i.e., the second-order information was rather useless.

[Final Decision · Program Chairs · 06 Feb 2017]
**ICLR committee final decision**

The paper proposes an interesting approach, in that (unlike many second-order methods) SR1 updates can potentially take advantage of negative curvature in the Hessian. However, all reviewers had some significant concerns about the utility of the method. In particular, reviewers were concerned that the method does not show a significant gain over the Adam algorithm (which is simpler/cheaper and easier to implement). The public reviewer also points out that there are many existing quasi-Newton methods designed for DL, so it is up to the authors to compared to at least one of these. For these reasons I'm recommending rejection at this time.